# Contributions of Investment and Employment to the Agricultural GDP Growth in Egypt: An ARDL Approach

Nouran Abdelhamid Abdelgawwad [1] and Abdelmonem Lotfy Mohamed Kamal [2,*]

[1] Department of International Trade, Faculty of International Logistics and Transportation, Arab Academy for Science and Technology (AAST), Cairo 1179, Egypt; nabdelhamid@aast.edu

[2] Department of Finance and Investment, Faculty of Business Administration, Economics, and Information System, Misr University for Science and Technology (MUST), 6th of October City, Giza 11556, Egypt

[*] Correspondence: abdelmonem.lotfy@must.edu.eg

**Abstract:** This paper explores the impact of investment and employment on Egypt's agricultural growth during the period 1991 to 2021 using annual time series data. We use the ARDL approach to examine the long-run and short-run relationships among agricultural investment, agricultural employment and agricultural GDP. The results reveal that the variables of interest are bound together in the long run. The long-run relationship and the error correction model are estimated. The accompanying equilibrium correction proves that long-run linkages exist in a meaningful way. Results show that agricultural investment and agricultural employment are major short- and long-run determinants of the agricultural GDP. In the long run, every 1% increase in agricultural employment (AEMP) results in an increase in the agricultural GDP (AGDP) of 3.73%, while every 1% increase in agricultural investment (AINV) improves the AGDP by 0.43%. In the short run, 26% of all disequilibrium-causing motions are adjusted for in a single session. Therefore, it takes 3.85 years for the Egyptian agricultural GDP to achieve the transition from a short-term disequilibrium situation to a long-term equilibrium. Thus, decision makers should increase the rates of investment in the agricultural sector, in parallel to the development of the agricultural labor force in Egypt. Moreover, the increased allocation of public investments and the injection of private investments are highly recommended. In addition, the Egyptian agricultural sector needs improvements regarding human capital development and agricultural training. Finally, the government must initiate comprehensive farmer support services, bolstered farm/non-farm links and the promotion of rural SMEs to serve as the foundation for agricultural and rural development.

**Keywords:** ARDL approach; agricultural investment; agricultural employment; Egypt's agricultural GDP; bounds test





## 1. Introduction

Agriculture is the most important sector of the Egyptian economy because of its contribution to food security and job creation. In recent years, there have been serious economic concerns about reducing food imports and the management of the foreign debt of Egypt (Goueli and Miniawy 1994). In the early 1970s, the Egyptian agricultural sector contributed approximately 28% of the total GDP but employed approximately 53% of the workforce.

By the end of the 1980s, the contribution of the agricultural sector to the GDP had declined to approximately 17% and its share in total employment had dropped by nearly one third. Today, the agricultural sector contributes approximately 13.6% of the gross domestic product and provides approximately 29.2% of all jobs, and the value of agricultural exports represents only approximately 11.5% of the total value of Egyptian exports (CAPMAS 2022), which implies that the role of the agricultural sector in Egypt is gradually decreasing. Egyptian agriculture is characterized by the domination of small farms and traditional practices, which need a few alterations to meet international standards (USAID 2022).

Egypt's government has implemented a number of economic reforms since 1986 in an effort to lessen both internal and foreign imbalances, remove economic distortions and encourage sustainable growth in the productive sectors. Policymakers have also taken action to ease constraints on the agricultural industry. Pricing and marketing restrictions on agricultural products have been scaled back or eliminated (Goueli and Miniawy 1994).

For governments seeking to improve their own domestic products, a good starting point is with realistic evaluations of the possibilities for employment and investment. Given the current economic and political climate, agricultural investment is crucial to Egypt's national economy. For the achievement of targeted agricultural development in Egypt, the gross and agricultural domestic products, agricultural income, the share of the agricultural sector in the gross domestic product, the annual growth rate of the agricultural sector and the per capita share of the gross agricultural product are all significantly impacted by agricultural investment (Hassan et al. 2016). This primarily entails taking into account the prospective effects on employment, both direct and indirect, as well as the corresponding socioeconomic concerns and potential adverse effects on livelihoods.

To accommodate population growth, the agricultural production of Egypt should be widely expanded. To achieve this goal, the government strives to pursue two main strategies as a priority: the reclamation of vast desert areas and the intensive cultivation of the existing arable land using high-tech practices of farming (El-Khalifa and Zahran 2022). To implement these strategies, the Egyptian agricultural sector is expected to need large capital. Thus, investment in the agricultural sector becomes an urgent necessity.

Agricultural investment is an important tool for economic development and output growth in Egypt (Kamal and AboElsoud 2023). Apart from its importance in absorbing human resources and reducing unemployment rates, it is key to increase the total factor productivity (TFP). Furthermore, agricultural investment is essential to increase agricultural exports and improve the trade balance, which give rise to considerable national income. On the other hand, agricultural employment is considered a major determinant of agricultural development in Egypt. It is one of the most important inputs that policymakers should keep in mind when planning to increase agricultural production (Singhal et al. 2023).

As agricultural production is characterized by seasonality, which negatively affects the agricultural labor market (Christiaensen and Maertens 2022), it is essentially relies on the volume of agricultural employment, besides the prospective role of agricultural investment. Therefore, this paper examines the interactive impact of investment and employment on the agricultural GDP, as an indicator of agricultural growth, in Egypt, which may guide policymakers to predict and prepare for the long-run effects of investment and employment on Egyptian agriculture. Thus, the basic research questions that this article seeks to address are as follows: (1) What is the nature of the relationship among agricultural investment, agricultural employment and the agricultural GDP in Egypt? (2) Does agricultural investment have any effect on the Egyptian agricultural GDP in the short or the long run? (3) Does agricultural employment have any effect on the Egyptian agricultural GDP in the short or the long run?

Consequently, this study broadly aims to evaluate the role of both agricultural investment and agricultural employment in the growth of the agricultural sector in Egypt under the period of evaluation, 1991 to 2021.

Moreover, the present study stems from the fact that identifying the relationship of Egyptian agricultural domestic products with their most important determinants (investment and employment) is considered the first building block in planning to develop the agricultural sector in Egypt. Besides policymakers, the results undoubtedly will also help decision makers in scheduling the implementation of programs, plans and policies. Furthermore, this paper adds to the literature in the discipline of study.

Therefore, the present study attempts to fulfil several main objectives. The first is to identify whether there is a co-integration relationship among the study variables (agricultural investment, agricultural employment and agricultural GDP); the second is to examine the long-run effects of agricultural investment and agricultural employment

on the agricultural GDP; and the third is to determine the short-run effects of agricultural investment and agricultural employment on the agricultural GDP.

The rest of the paper is composed of four sections. The second section is devoted to the literature review. The third describes the methodology. The fourth presents the findings. The fifth concludes the article and provides recommendations.

*Literature Review*

The literature provides conflicting approaches to analyzing the relationships of both agricultural investment and agricultural employment and the agricultural GDP. The vast majority of these approaches are not related to Autoregressive Distributed Lag (ARDL).

In recent years, several studies have focused on the factors affecting agricultural growth in various countries. First of all, it is important to mention that none of the previous studies have combined agricultural employment and agricultural investment (as independent variables) to determine their interactive impacts on the agricultural GDP (the dependent variable) as a representative of agricultural growth.

In light of the recent political and economic climate, as well as the state's low domestic production, Ismail (2020) aimed to investigate the effectiveness of agricultural investment and its significance for Egypt's national economy. To do this, the study examined the development of both national and agricultural investments. It employed two variables to perform the study for Egypt: national and agricultural investments. The study's findings indicated that the indices of agricultural investment efficiency confirmed this efficiency by increasing returns on investment and investment multipliers greater than one. Moreover, the low endemism coefficient of agricultural investment, which measured how much the agricultural sector contributed to the GDP and was less than one, indicated that the sector did not receive investments that were proportionate to its GDP-generating contribution. Consequently, the present paper reviews these points of research to investigate how investment and employment contribute to the growth of the agricultural sector in Egypt.

According to Vuppalapati (2021), a large population and labor force, as is the case in Egypt, has enormous prospects for growth and progress thanks to technical advancements in agriculture. Thus, the study used the variables of employment and artificial intelligence (AI) to test the contribution of the agricultural sector to GDP growth in the developing countries of Brazil and India. Any increase in agricultural worker productivity was found to be impactful and could solve both the issues encountered by agriculture and the emerging demands, since the model showed a strong correlation with employment in the sector and the general population. The present paper seeks to apply some of these implications in the Egyptian agricultural sector.

Kotb (2017) identified the most important factors affecting the agricultural GDP during the period 2015–2020 and then ranked these factors according to their relative importance in influencing the agricultural GDP. To achieve this goal, the study used the Ordinary Least Squares (OLS) method. Furthermore, it depended on the estimations of simple regression equations to analyze the mono-relationships between the dependent variable (agricultural GDP) and each of the independent variables (the crop area, the amount of agricultural employment, the number of animal units, the number of agricultural tractors, the value of agricultural investment and the value of agricultural loans). The results of the study confirmed that both agricultural employment and agricultural investment had a statistically significant positive effect on the agricultural GDP. However, according to their relative importance in influencing the agricultural GDP, agricultural investment occupied fifth position, while agricultural employment occupied sixth position.

Another example of a study that did not use the ARDL is Mohamed and Ahmed (2019), which followed Bakari and Mabrouki (2018). It measured the impact of agricultural investment on agricultural development (represented by the agricultural GDP) in Egypt. The Two-Stage Least Squares method (2SLS) was used to devise a simultaneous equations model. Thus, the study did not use ARDL or the bounds test for co-integration, concentrating only on the short-run relationship. However, the results of the study con-

firmed that agricultural investment had a statistically significant and positive impact on the agricultural GDP.

Moreover, Abdelaal and El-Shafei (2022) investigated the main factors affecting the value of the agricultural output in Egypt during the period 1991–2019. This study depended on OLS, which means that it did not consider long-run effects. Besides this, the study used the productivity of the agricultural workers to estimate the effect of agricultural employment on the value of the Egyptian agricultural output. Nevertheless, the results showed that both employment and investment had a significant positive impact on the value of the agricultural output.

In contract, El-Rasoul (2018) and Appleton and Balihuta (1996) incorporated new variables that play a vital role in the agricultural sector. The two studies investigated the role of agricultural education on the growth of the agricultural sector in Egypt. Additionally, they employed Johansen and Juselius (1988) and Johansen and Juselius (1990)'s approach of co-integration, which differs from ARDL in relation to the conditions of use. The results of these two studies indicated that there is a one-way causal relationship that extends from agricultural education to the agricultural GDP.

Al-Okl (2023) analyzed the impact of green support (support for services, infrastructure, education and training) on the agricultural GDP of some developed countries (European Union, United States and Japan) using the ARDL model. Despite the importance of this study, it did not consider agricultural investment or agricultural employment as the most important determinants of the agricultural GDP.

Notably, the results of research on the empirical relations among the variables under consideration vary greatly in accordance with the model of analysis. Therefore, the debate regarding the impacts of investment and employment on the growth of the agricultural sector is still ongoing and remains open to further study. As a supplementary contribution to the literature in this discipline, this paper examines the short- and long-run impacts of investment and employment on the agricultural GDP in Egypt using the ARDL approach.

## 2. Methodology

### 2.1. Description of the Study Area

Egypt, officially the Arab Republic of Egypt, is the largest country in the Arab World in terms of population. It is located in the northeastern part of the African continent, extending into the Sinai region across the Gulf of Suez. Egypt has a total size of 1,002,000 square kilometers, approximately 96.5% of the total area is desert land, and only approximately 3.5% of this is under cultivation. The total length of the Egyptian coast is 2936 km, of which only 995 km is along the Mediterranean, while 1941 km is along the Red Sea (Goueli and Miniawy 1994).

The Egyptian economy depends mainly on agriculture, as well as tourism, the Suez Canal, media and natural gas exports. In addition, there are approximately 3 million Egyptians working abroad, mainly in Saudi Arabia, the United Arab Emirates, Kuwait, Libya and Europe. When the Aswan High Dam and Lake Nasser were completed, the importance of the Nile River in the agriculture of Egypt was significantly altered. The limited arable land and the dependence on the Nile, as well as the rapidly growing population, all served to overtax resources and stress the Egyptian economy (Wikipedia 2021).

In this study, the following hypotheses are tested using three variables: agricultural investment, agricultural employment and the agricultural GDP.

**Hypothesis 1.** *There is a co-integration relationship among the variables (agricultural investment, agricultural employment and agricultural GDP).*

**Hypothesis 2.** *There is a statistically significant positive effect of agricultural investment on the Egyptian agricultural GDP in the short and long term.*

**Hypothesis 3.** *There is a statistically significant positive effect of agricultural employment on the Egyptian agricultural GDP in the short and long term.*

The three variables employed are illustrated in the following Figures 1–3[1]. Data is obtained from The Financial Monthly Report published by the Egyptian Ministry of Finance (2022) and Egyptian National Accounts Data (2022) published by the Egyptian Ministry of Planning and Economic Development. Also, World Development Indicators published online by the World Bank (2023) is used for employment data.

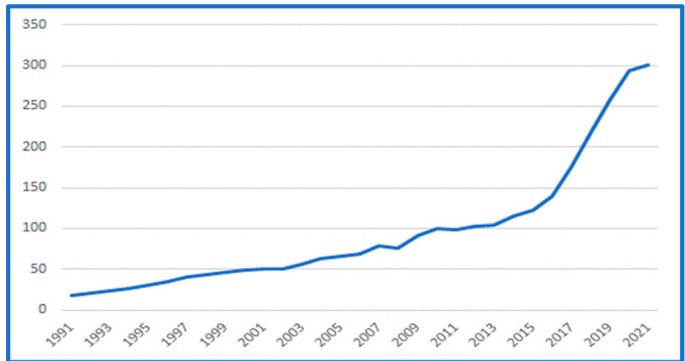

**Figure 1.** Output in Agriculture Sector in Egypt through 1991 to 2021 (in billion LE).

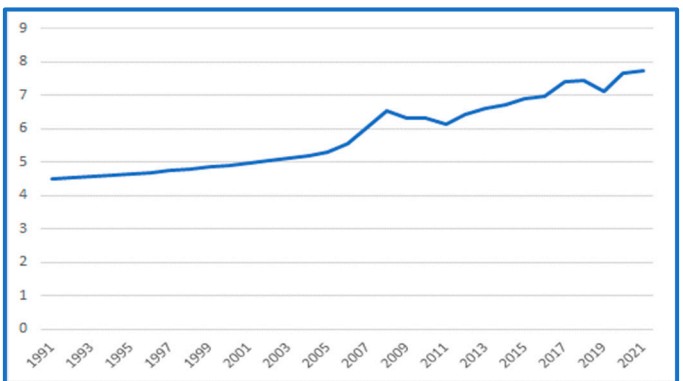

**Figure 2.** Employment in Agriculture Sector in Egypt through 1991 to 2021 (in million).

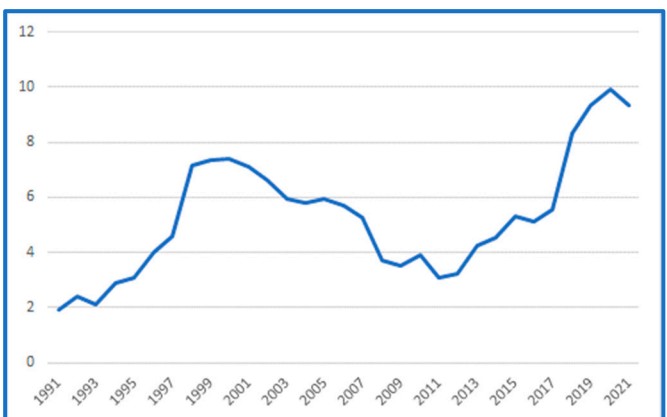

**Figure 3.** Investments in Agriculture Sector in Egypt through 1991 to 2021 (in billion LE).

*2.2. Data Collection*

The study mainly employed annual data that covered a period of 31 years (1991–2021) for Egypt. The data were collected from three sources: the Ministry of Agriculture and Land

Reclamation (MALR), the Central Agency for Public Mobilization and Statistics (CAPMAS) and the Ministry of Planning and Economic Development (MPED).

### 2.3. Model Specification

We followed the unconstrained error correction model (ECM) that implements the ARDL technique within the framework of Pesaran (1997) and Pesaran et al. (2001). The ARDL model:

$$\Delta LAGDP = \beta_o + \beta_1 LAINV_{t-1} + \beta_2 LAEMP_{t-1} + \sum \alpha_1 \Delta LAINV_{t-i} + \sum \alpha_2 \Delta LAEMP_{t-i} + \mu_t \quad (1)$$

By estimating this model using the Ordinary Least Squares (OLS) approach, the ARDL bounds test was applied. In order to perform a Wald test (F-statistic), limitations were placed on the projected long-run coefficients in this study. Thus, this test took the following form.

**Hypothesis 4.** *β1 = β2 = 0 (no long-run relationship).*

**Hypothesis 5.** *β1 ≠ β2 ≠ 0 (a long-run relationship exists).*

In order to obtain a conclusion, the estimated F-statistic value was then contrasted with the crucial values shown in the Pesaran et al. (2001) table.

The second step after establishing co-integration is to choose the ARDL model lags' ordering using either AIC or SBC. According to Pesaran and Pesaran (1997), the best lag order is determined by the smallest AIC or SBC value. As a result, the conditional ARDL long-run model can be obtained as follows:

$$LAGDP = \beta_o + \sum \beta_1 LAINV_{t-i} + \sum \beta_2 LNAEMP_{t-NA} + \mu_t \quad (2)$$

The last step is to calculate the short-run dynamic parameters using the long-run estimates to calculate the ECM. The following estimation is possible:

$$\Delta LAGDP = \beta_o + \Omega ECM_{t-1} + \sum \alpha_1 \Delta LAINV_{t-i} + \sum \alpha_2 \Delta LAEMP_{t-i} + \mu_t \quad (3)$$

The short-run dynamic coefficients in this model are $\alpha1$ and $\alpha2$, while the speed of the adjustment parameter is $\Omega$. Additionally, by applying Equations (1) and (2) to forecast the equilibrium relationship, the error correction term ECM can be calculated as described in Equation (3).

Thus, the study examined stationarity before evaluating the long-run relationships to ensure that the integrated order of the studied series was appropriate for the use of the ARDL approach. As a result, the study used the Augmented Dickey–Fuller (ADF) and Phillips–Perron (PP) tests as its two unit root tests. The existence of a unit root is the null hypothesis for these two tests, while the alternative hypothesis is that no unit root exists.

In light of the preliminary analysis, it was found that only two of the time series of the study variables, namely investment (AINV) and employment (AEMP), were not stationary at the level but stationary at the first difference. Meanwhile, the other, which was Egypt's agricultural GDP (AGDP), was stationary at the level. Accordingly, it was not possible to use OLS. In such cases, the Autoregressive Distributed Lag (ARDL) approach is the best choice, because of the spurious regression that results from the use of OLS (Nkoro and Uko 2016).

To the best of our knowledge, no study has been previously conducted on this topic in Egypt using ARDL approach. According to the comprehensive literature review, most of the previous studies related to the study variables have focused on analyzing the short-run impacts. In this study, therefore, the ARDL model was used to highlight both the short-run and long-run effects of investment and employment on Egypt's agricultural GDP. In addition, we used the E-Views package. It is somewhat unique in the field of ARDL and offers a flexible and easy-to-use interface.

As for the definition of ARDL, in sum, ARDL is an analytical model that is particularly used to investigate the co-integration relationships of variables. It is considered a solution for the spurious regression problems of OLS. Moreover, ARDL is an effective and accurate tool to plan for the long run. In this study, firstly, we demonstrate step by step the procedure of the implementation of ARDL and its components and tests. Methodically, we followed the following steps.

- Examining the data (descriptive statistics—graphs).
- Ensuring that all variables were integrated of order I(0) or I(1), or a combination of them. To do this, the Augmented Dickey–Fuller (ADF) ((Dickey and Fuller 1979), (Dickey and Fuller 1981), (Johansen 1991), (Cheung and Lai 1995), (Elliott et al. 1996)) and Phillips–Perron (P.P.) (Phillips and Perron 1988) tests were used.
- Determining the appropriate lag structure of the selected model. For this purpose, the study followed the process of E-Views to select the optimal lags based on the evidence from several criteria (Akaike (AIC), Schwarz (SH), Hannan–Quinn (HQ), Final Prediction Error (FPE) and sequential modified LR test statistic).
- Estimating the model (ARDL model) (Sarkodie and Owusu 2020).
- Diagnostic tests. Diagnostic tests were conducted to ensure that the model was free from serial correlation and heteroskedasticity. As for the serial correlation, the Breusch–Godfrey serial correlation LM test was used to verify that the residuals from the model were serially uncorrelated. In regard to the residual homoskedasticity, the Breusch–Pagan–Godfrey test was chosen for this purpose.
- Stability tests. This stage of analysis is very important to examine the stability of the long-run coefficients. The E-Views package offers several tests for stability diagnostics. In this study, we chose the CUSUM and CUSUM squares tests, which Brown et al. (1975) developed. In accordance with both tests, the structural stability of the estimators is proven only if their statistic graphs fall within the critical limits at the significance level of 5% ((Ali 2017), (Durmaz and Lee 2015), (Dritsakis 2011), (Halicioglu 2007), (Shahrestani and Sharifi-Renani 2007)).
- Performing the bounds test and estimating the long- and short-run coefficients and the speed of adjustment (ARDL Long-Run Form and Bounds Test—ARDL Error Correction Regression).

## 3. Results

### 3.1. Descriptive Statistics

Table 1 provides the descriptive statistics on the raw data of the study variables over the study period (1991–2021). From this table, we can see that the mean value of LAGDP is 95.69, the median is 69.38, the maximum value is 300.52 and the minimum value is 18.61. Skewness is not zero. Thus, the data are not normally distributed. In the case of LAINV, the mean is 3.31, the median is 5.26, the maximum value is 9.91 and the minimum value is 1.94. Skewness is not zero, so the data on agricultural investment are not normally distributed. The LAEMP mean is 5.82, the median is 5.45, the maximum level of agricultural employment is 7.73 and the minimum level is 4.51. Skewness is not zero, so the data are not normally distributed. Regarding kurtosis, the values 1.664, −0.523 and −1.369 for LAGDP, LAINV and LAEMP, respectively, give support to the skewness results, confirming that the data are not normally distributed.

**Table 1.** Descriptive statistics and correlations of the study variables during 1991–2021.

|  | LAGDP | LAINV | LAEMP |
|---|---|---|---|
| Mean | 95.69 | 5.31 | 5.82 |
| Median | 69.38 | 5.26 | 5.45 |
| Maximum | 300.52 | 9.91 | 7.73 |
| Minimum | 18.61 | 1.94 | 4.51 |
| Std. Dev | 77.7 | 2.18 | 1.08 |
| Skewness | 1.535 | 0.459 | 0.343 |
| Kurtosis | 1.664 | −0.523 | −1.369 |
| Shapiro–Francia W test for normality | 5.671 | 6.416 | 5.323 |
| Probability | 0.00001 | 0.00001 | 0.00001 |
| Shapiro–Wilk W test for normality | 6.135 | 6.967 | 5.755 |
| Probability | 0.00000 | 0.00000 | 0.00000 |
| Skewness/kurtosis tests for normality | 23.86 | 55.69 | 24.50 |
| Probability | 0.00000 | 0.00000 | 0.00000 |
| Obs. | 31 | 31 | 31 |
| LADGP | 1.000 |  |  |
| LAINV | 0.3272 * | 1.000 |  |
| LAEMP | 0.4705 * | 0.9848 * | 1.000 |

* Denotes significance at a 95% confidence level. Variable definition: LAGDP, LAINV and LAEMP are the logarithmic forms of Egypt's agricultural GDP, investment, capital and employment, respectively. Source: Authors' calculations using E-Views 12.

### 3.2. Stationary Tests (Unit Root Tests)

One of the conditions of using ARDL is the integration order of variables. It is necessary before applying ARDL to ensure that all variables are integrated of order I(0), I(1) or a combination of them. In other words, we must ensure that no series is integrated of order 2 or higher. In this test, the null hypothesis assumes that the series has a unit root (it is not stationary), while the alternate hypothesis assumes that there is no unit root (it is stationary). Tables 2 and 3 illustrate the results of the ADF and P.P. tests for the log of each series at the level and the first difference, including the three cases (intercept, trend and intercept and none) at the significance level of 5%.

**Table 2.** Results of unit root tests via Augmented Dickey–Fuller Test (ADF).

|  | Level | | | First Difference | | | |
|---|---|---|---|---|---|---|---|
| Variable | t-Value | Critical Value | *p*-Value | t-Value | Critical Value | *p*-Value | Level of Integration |
| LAGDP | −3.76 | −3.57 | 0.034 ** | N/A | N/A | N/A | I(0) |
| LAINV | −1.63 | −3.57 | 0.755 | −4.42 | −3.57 | 0.007 ** | I(1) |
| LAEMP | −2.59 | −3.56 | 0.288 | −5.12 | −3.57 | 0.001 ** | I(1) |

***, ** and * denote significance at 99%, 95% and 90% confidence levels, respectively. All variables used in this study are expressed in the natural logarithmic form. Source: Authors' calculations using E-Views 12.

**Table 3.** Results of unit root tests via Phillips–Perron test (P.P.).

|  | Level | | | First Difference | | | |
|---|---|---|---|---|---|---|---|
| Variable | t-Value | Critical Value | *p*-Value | t-Value | Critical Value | *p*-Value | Level of Integration |
| LAGDP | −3.77 | −3.57 | 0.033 ** | N/A | N/A | N/A | I(0) |
| LAINV | −1.87 | −3.57 | 0.646 | −4.47 | −3.57 | 0.006 ** | I(1) |
| LAEMP | −2.44 | −3.56 | 0.351 | −5.61 | −3.57 | 0.000 ** | I(1) |

***, ** and * denote significance at 99%, 95% and 90% confidence levels, respectively. All variables used in this study are expressed in the natural logarithmic form. Source: Authors' calculations using E-Views 12.

From Table 2, we can see that the series of LAGDP does not have a unit root at the level and so it is I(0). On the other hand, LAINV and LAEMP have a unit root at the level, whereas neither of them has a unit root at the first difference, so these two series

are I(1). Thus, these results support the view that the vast majority of macroeconomic variables usually are not stationary at their levels but are stationary at their first differences ((Drebee and Abdul-Razak 2020), (Xiao 2001), (Gil-Alana and Robinson 1997), (Bierens and Guo 1993), (Kwiatkowski et al. 1992)). The results, illustrated in Table 3, also confirm this conclusion, so we can therefore proceed to the next step.

The results support the validity of using the ARDL approach to model the relationships between the series and assessing the co-integration relationships in this set of variables.

### 3.3. Specifying ARDL Lag Structure

After testing the integration orders and ensuring that none of the variables is integrated of order 2 or higher, it is necessary to select an appropriate number of lags for the ARDL model. As presented in Table 4, according to LR, FRE, AIC, SH, HQ, and FPE, the optimal lag order is chosen to be one. Based on these results, the model is estimated using only the first lag differences for each variable.

**Table 4.** Information criteria for lag order selection.

| Lag | LR | FPE | AIC | SC | HQ |
|---|---|---|---|---|---|
| 0 | NA | 3.310070 | −6.408108 | −6.262943 | −6.366305 |
| 1 | 145.0829 * | 9.110010 * | −12.31048 * | −11.72982 | −12.14327 * |
| 2 | 10.22971 | 1.10009 | −12.15658 | −11.14042 | −11.86396 |
| 3 | 14.77449 | 9.480010 | −12.38788 | −10.93603 | −11.96965 |
| 4 | 5.866535 | 1.430009 | −12.14664 | −10.25949 | −11.60321 |
| 5 | 9.125141 | 1.580009 | −12.36685 | −10.04421 | −11.69801 |

* denotes lag order selected by each criterion (LR, FPE, AIC, SC, HQ). All variables used in this study are expressed in the natural logarithmic form. Source: Authors' calculations using E-Views 12.

### 3.4. ARDL Model and Bounds Tests

In order to identify the co-integration relationships for the variables under consideration, we estimated the ARDL model under the case of a restricted constant and no trend (case 2) ((Sarkodie and Owusu 2020), (Arize 2017), (EAIB 2017), (Eshraghi et al. 2011), (Pesaran et al. 2001), (Pesaran and Shin 1999), (Granger and Lin 1995), (Schimmelpfennig and Thirtle 1994), (Inder 1993), (Granger 1988), (Engle and Granger 1987), (Granger 1981), (Granger and Newbold 1974), (Granger 1969)). According to Pesaran et al. (2001), it is recommended to use case 2, since the results presented in Figures 1–3 did not show any noticeable trend and it is preferable to include the intercept in the long-run analysis. A restricted constant means that the intercept participates in the long-run relationships, which would be more suitable for the studied variables. In the present model, the dependent variable is LAGDP, while the independent variables (explanatory variables) are LAINV and LAEMP. The ARDL selected model is (0, 1, 1). Based on the bounds test, reported in Table 5, it is clearly shown that the F-statistic value of 18.341 is greater than the value of either the I(0) and I(1) F-bounds at all the significance levels. Thus, we reject the null hypothesis that states that there is no co-integration between LAGDP, LAINV and LAEMP. Accordingly, there is a long-run equilibrating relationship among the study variables.

**Table 5.** F-bound test for the selected ARDL model.

| F-Statistic | Significance Level | I(0) | I(1) |
|---|---|---|---|
| 18.34 | 10% | 2.63 | 3.35 |
| | 5% | 3.1 | 3.87 |
| | 2.5% | 3.55 | 4.38 |
| | 1% | 4.13 | 5.00 |

The null hypothesis: no level relationship. Source: Authors' calculations using E-Views 12.

## 4. Discussion

The results of the long-run relationship estimates are presented in Table 6. According to the T-statistic values and *p*-values (prob.), all long-run (equilibrium) coefficients are statistically significant, and their signs are consistent with the economic theory. In light of these results, agricultural investment (AINV) has a significant and positive impact on the agricultural GDP. An increase in agricultural investment by 1% improves the agricultural GDP by 0.43% in the long run. Furthermore, the results also indicate that agricultural employment (AEMP) is an important determinant of the agricultural GDP. Every 1% increase in agricultural employment yields an increase of 3.73% in the agricultural GDP in the long run.

**Table 6.** Estimates of long-run coefficients using the selected ARDL model.

| Variable | Coefficient | T-Statistic | Prob. |
|---|---|---|---|
| C | −1.14 | −6.66 | 0.0005 |
| LAINV | 0.43 | 4.04 | 0.0000 |
| LAEMP | 3.74 | 14.22 | 0.0000 |

Source: Authors' calculations using E-Views 12.

According to World Bank[2], the agriculture value added per worker in Egypt in 1991 was USD 3122.78 and it reached USD 6096.88 in 2017, calculated according to 2015 USD. This indicator, also in 2017, reached USD 7745.36 in the MENA region and USD 37,933.34 in the high-income group of countries, whereas it was USD 4160.44 and USD 14,739.5 in 1991, respectively.

Therefore, it is concluded that the employment contribution to the agricultural GDP in Egypt is below the average of similar countries in the Middle East and very weak compared to the high-income countries[3]. Of course, these findings are consistent with both the past and present literature. In the past literature, one can find (Johnston and Mellor 1961), (Kuznets 1961), (Charles 1965) and (Witt 1965), whose studies considered the issues of overpopulation, agricultural underemployment and zero marginal productivity of labor. This body of literature considered the situation in Egypt, where the challenge is to promote agricultural outputs under the conditions of two limitations: overpopulation and underemployment. As a result, other inputs must be developed to replace (unavailable additional) land and likely some labor. The effective use of such inputs necessitates both capital and labor force modifications in the labor management complex (Witt 1965).

With regard to the recent literature, the findings of the present paper match those of Nyiwul and Koirala (2022), which stated that nations with overall less favorable investment environments suffer from low levels of agricultural outputs. These nations, such as Egypt, can enhance their agricultural outputs by attracting foreign direct investment (FDI) and boosting investment levels. It has been proven that FDI and productive investments result in better technologies, technical expertise, practices, management and other systems that benefit the host nation.

On the other hand, Kheir-El-Din and Moursi (2006) argued that the apparent deficiency in investment could be explained by the Egyptian economy's potential overshoot of the capital–labor ratio. They concluded that higher levels of investment and capital are therefore not "….always conducive for growth". Unlike Kheir-El-Din and Moursi 2006, the present paper refutes this argument via several claims. First, if and only if other inputs are held constant, it is believed that applying the rule of diminishing returns to capital is correct. This is not the case with the Egyptian economy, however, as other production factors, such as employment in the agriculture sector, have been rising over time, as shown in Figure 2. Secondly, and most urgently, as Kamal and AboElsoud (2023) illustrated, there has been a sharp decline in the capital growth ratio in the Egyptian economy in the period of 1990 to 2019, while the outputs and the capital–labor ratio have been both rising despite the savings rate declining. Therefore, the apparent weak contribution of investment and employment in Egypt's agricultural GDP could be attributed to the lack of capital accumulation brought

about by a dramatic fall in savings (Kamal and AboElsoud 2023). Consequently, the present paper argues that the agricultural sector in Egypt is in need of more capital accumulation via boosting both fiscal and human capital. Thus, it is crucial to invest heavily in farmers to enable them to become skilled laborers.

The results of the short-run coefficients associated with the long-run relationships are given in Table 7. The estimated value of R-squared is 0.412, which means that approximately 41.2% of the variation in the agricultural GDP (in the short run) is attributed to the independent variables included in the model (AINV and AEMP). The signs of the short-run impacts differ from their long-run counterparts. Moreover, the coefficient of agricultural investment difference D(LAINV) is statistically significant, while it is not significant for agricultural employment D(LAEMP); however, such impacts usually are observed only in the short run. Accordingly, we conclude that variables that have positive long-run impacts may not have significant impacts in the short run, and vice versa. Importantly, the error correction (EC), which is referred to in Table 8 as CointEq(−1), is negative, with a coefficient estimate of 0.26.

**Table 7.** Short-run coefficients and error correction for the selected ARDL model.

| Variable | Coefficient | T-Statistic | Prob. |
|---|---|---|---|
| D(LAINV) | 0.21 | 3.88 | 0.0007 |
| D(LAEMP) | 0.24 | 0.70 | 0.4889 |
| CointEq(-1) | −0.26 | −7.89 | 0.0000 |
| R-squared | 0.412 | | |
| Adjusted R-squared | 0.364 | | |
| Durbin–Watson stat. | 1.92 | | |

Source: Authors' calculations using E-Views 12.

**Table 8.** Results of residual diagnostics (serial correlation and heteroskedasticity) and functional form.

| Test | F-Statistic | *p*-Value |
|---|---|---|
| Ramsey's RESET Test * | 32.55 | 0.0000 |
| Breusch–Godfrey Serial Correlation LM ** | 0.2189 | 0.8051 |
| Heteroskedasticity test: Breusch–Pagan–Godfrey *** | 1.61100 | 0.1506 |

* Null hypothesis: the model has no omitted variables; ** null hypothesis: no serial correlation; *** null hypothesis: homoskedasticity. Source: Authors' calculations using E-Views 12.

This implies that approximately 26% of any movements into disequilibrium are corrected for within one period. In other words, the period that is required to move from a disequilibrium situation (in the short run) to an equilibrium state (in the long run) is 3.85 years (1/0.26 = 3.85). Furthermore, the large value of the T-statistic (−7.89) implies that the coefficient of error correction is highly significant. This transitional period to equilibrium is large enough to permit the release of labor to other industries with higher productivity and more jobs. However, since there are significant unemployment rates in the Egyptian economy, this transfer may not stimulate overall economic growth. Therefore, more focus should be placed on the promotion of non-farm businesses, especially those that are connected to the agricultural sector, in addition to expanding to new lands and enticing rural labor to relocate there. In terms of creating jobs and revenue, a plan that emphasizes bolstering farm/non-farm links in historic lands is more likely to be successful.

In addition to the above, given the results reported in Tables 6 and 7, it can be clearly noticed that the impact of the independent variables in the long run is higher than their impact in the short run, which means that the impacts of the changes in agricultural employment and agricultural investment are much stronger in the long run.

### 4.1. Diagnostic Test

The results of residual diagnostics are given in Table 8. Since the null hypothesis of the Breusch–Godfrey serial correlation LM test is that the residuals are serially uncorrelated, the F-statistic *p*-value of 0.8051 indicates that we accept this null hypothesis, which implies that the residuals are serially uncorrelated. Similarly, since the null hypothesis of the Breusch–Pagan–Godfrey test is that the residuals are homoskedastic, the F-statistic *p*-value of 0.1506 indicates that we accept this null hypothesis; thus, the residuals are homoskedastic.

To test for the functional form, one quite useful approach to a general test for functional form misspecification is Ramsey's regression specification error test (RESET). Table 8 shows that we can accept the null hypothesis of Ramsey's RESET, which confirms that the model is specified well.

### 4.2. Stability Test

Testing the structural stability of the long-run coefficients, the results confirm that the estimated coefficients are structurally stable over the study period. As shown in Figures 4 and 5, both plots of the cumulative sum (CUSUM) and the cumulative sum of squares (CUSUMQS) fall within the critical limits at the significance level of 5%.

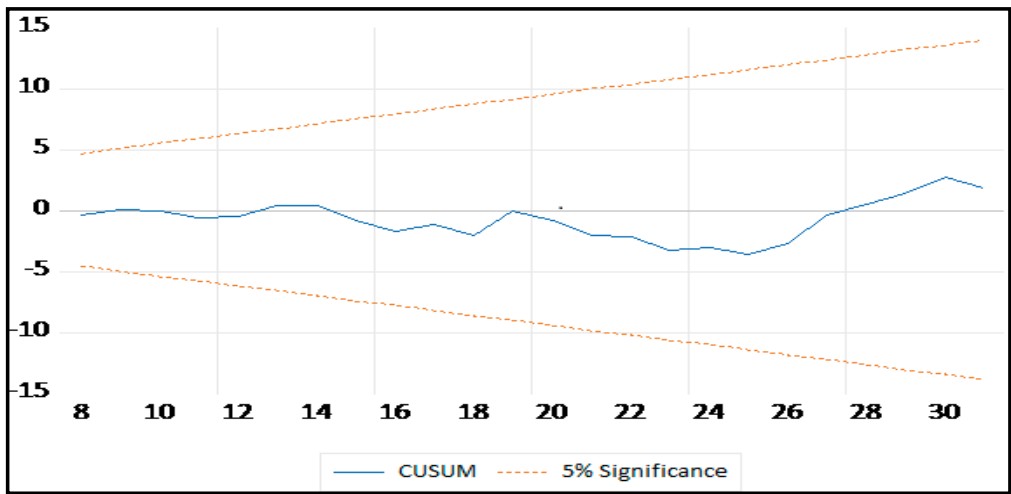

**Figure 4.** Plot of CUSUM test.

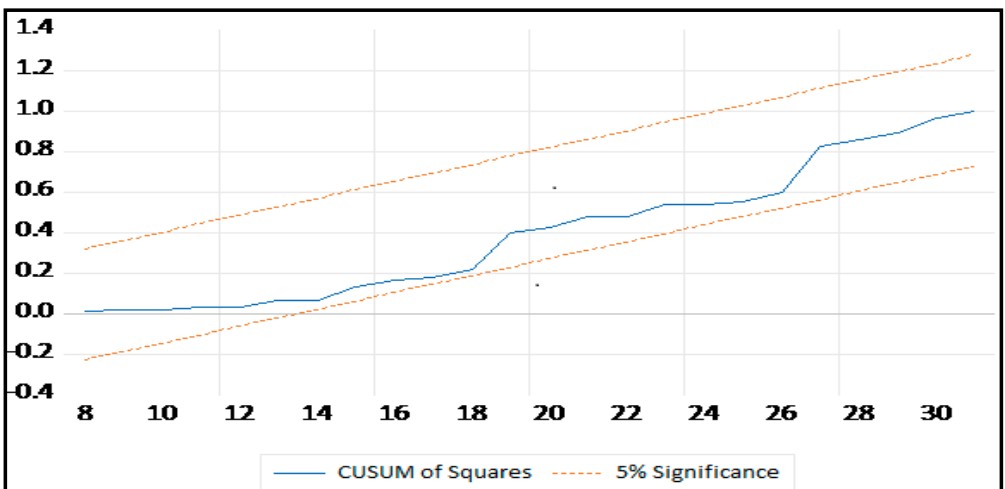

**Figure 5.** Plot of CUSUMSQ test.

Thus, given the results of the diagnostic and stability tests, it is evident that the estimated model does not have problems either with serial correlation or with heteroskedastic-

ity, and the stability of the coefficients is proven throughout the study period. Hence, we can confidently accept its estimators.

## 5. Conclusions

The question motivating this study was whether investment and employment can be targeted in order to develop the agricultural sector in Egypt. Thus, the study objective was to identify the relationships among agricultural investment, agricultural employment and the agricultural gross domestic product in Egypt during 1991 to 2021, using the Autoregressive Distributed Lag modeling approach to co-integration analysis. The ARDL bounds test and ECM regression provided evidence of a long- and short-run effect of both agricultural investment and agricultural employment on agricultural growth (represented by the agricultural gross domestic product). In the long run, every 1% increase in agricultural employment (AEMP) results in an increase in the agricultural GDP (AGDP) of 3.73%, while every 1% increase in agricultural investment (AINV) improves the AGDP by 0.43%. In the short run, approximately 41.2% of the variation in the AGDP is attributed to the model's regressors.

The equilibrium error correction coefficient (–0.26) implies a high speed of adjustment. Additionally, the error correction coefficient has a negative sign and it is also highly significant, which confirms the existence of a long-run relationship. It implies a high speed of adjustment, as approximately 26% of the disequilibrium in the short run converges back to the long-run equilibrium in one year. Consequently, this implies that the Egyptian agricultural GDP takes 3 years and 10 months to achieve a transition from short-term disequilibrium to long-term equilibrium. This equilibrium transition period is long enough to allow for the transfer of labor to other sectors with better productivity and jobs. However, this transfer might not promote general economic growth given the high unemployment rates in the Egyptian economy. In addition to expanding to new lands and encouraging the rural workforce to relocate there, emphasis should be placed on the promotion of non-farm enterprises, especially those that are linked to the agricultural industry. A strategy that prioritizes strengthening farm/non-farm ties in historic lands is more likely to be effective in terms of generating jobs and money.

The results also show that the estimated model does not have problems with serial correlation and heteroskedasticity. Furthermore, the stability of the long-run coefficients is proven throughout the study period. On the basis of these results, it is clear that agricultural investment and agricultural employment play a key role in the growth of the Egyptian agricultural sector. The study therefore recommends that more public investment should be allocated to the agricultural sector, in order to expand the farming land and support the agricultural inputs. Increases in private sector investment are also required, especially in the area of mechanization and livestock. Moreover, human capital development and agricultural training should be given more attention. It is of the utmost importance to strengthen the agricultural extension system in order to keep farmers informed and aware of the most relevant practices and innovations needed to increase agricultural productivity.

Land, information and financial constraints, as well as inadequate transportation and storage infrastructure, limit agricultural investments and employment. Through investments and public policy, these restrictions must be loosened. As stated previously, it is crucial to invest heavily in farmers and infrastructure in the Egyptian agricultural sector. However, restricted cropped areas are unlikely to employ the entire rural labor force due to the Nile Valley and the Delta's restrictions on the availability of old lands. To utilize the labor force that is available and offer them sufficient sources of income, other non-agricultural activities must be created. Global experience indicates that the three primary pillars of comprehensive farmer support services, bolstered farm/non-farm links and the promotion of rural SMEs serve as the foundation for agricultural and rural development. In fostering such pillars, the government must take the lead.

Regarding future research directions, we aim to extend this analysis in the near future by exploring the topic of "agricultural productivity growth and poverty alleviation in Egypt".

**Author Contributions:** Conceptualization, N.A.A. and A.L.M.K.; Methodology, A.L.M.K.; Investigation, A.L.M.K.; Resources, N.A.A.; Data curation, N.A.A.; Writing—original draft, N.A.A.; Writing—review & editing, A.L.M.K. All authors have read and agreed to the published version of the manuscript.

**Funding:** The APC was funded by Misr University for Science and Technology, Grant no. 5.

**Informed Consent Statement:** Not applicable.

**Data Availability Statement:** The data used is open-source. Neither classified nor trade secret data used. The data is available at https://www.capmas.gov.eg/Pages/StaticPagesaspx?page_id=5034 (accessed on 10 March 2023), https://mof.gov.eg/en (accessed on 12 March 2023), and https://mped.gov.eg/Analytics?lang=en (accessed on 15 March 2023).

**Conflicts of Interest:** The authors declare no conflict of interest.

## Notes

1.   Egyptian Ministry of Finance. The Financial Monthly Report, December 2022. Available online: https://mof.gov.eg/en (accessed on 12 March 2023). World Bank, World Development Indicators. Available online: https://data.worldbank.org/indicator/ (accessed on 12 March 2023).
2.   Source: World Bank based on data from multiple sources at https://ourworldindata.org/employment-in-agriculture (accessed on 12 May 2023).
3.   Agriculture value added per worker is calculated as the total amount of economic value generated from farming divided by the number of people employed in agriculture.

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
