# Peer review of "Contributions of Investment and Employment to the Agricultural GDP Growth in Egypt: An ARDL Approach"

_economies, doi:10.3390/economies11080215_

Round 1

Reviewer 1 Report

This is an interesting paper about development of agricultural sector in Egypt while considering investment and employment as main accelerators of growth. There has been applied Autoregressive Distributed Lag (ARDL) models to determine the short-run and long-run effects of agricultural investment and agricultural employment on agricultural GDP in Egypt.

The title well presents the paper’s concept.

The Abstract is informative, can stand alone, and covers the combination of the problem and obtained results. Some recommendations would be advisable to be added in this part too.

Key words are correctly described.

The introduction covers information about the problem statement and gives a good background for further research. The quoted literature is fitting well into the problem. The objectives are well addressed although not well fit in the structure of the paper, as are presented in the section 2.1 Description of the Study Area. This should be necessarily corrected.

There is more structurall errors – it is not know what is the puropse of section „2.3 The three variables are also illustrated in the following three figures” which should be better related to overall reasoning of the paper. This needs to be corrected.

In the section „Literature review” it is written that „In recent years, several studies have focused on the factors affecting the agricultural growth in various countries. First of all, it is important to mention that none of the previous studies have combined particularly between agricultural employment and agricultural in vestment (as independent variables) to determine their interactive impacts on agricultural GDP (the dependent variable) as a representative of agricultural growth”. Such statement needs to be elaborated more showing exact citations what factors are considered in previous stusies and in which countries (developed, developing). Additionally it should be discussed why in previous studies no one have combined agricultural employment and agricultural in vestment (as independent variables) to determine their interactive impacts on agricultural GDP (the dependent variable).

Materials and methods are well justified and presented. The ARDL method is the correct one for such a study. Data and data analysis are sufficiently described.

On page 3 – lines 149-151  it is stated „(Al-Okl 2023) analyzed the impact of green support (support for services, infrastructure, education and training) on the agricultural GDP of some developing countries (European Union, United States and Japan) using the ARDL model.” As the countries such EU, US, Japan are concerned as develop-ed countries.

The results are clearly presented and their significance is highlighted. The Autors explained and interpreter the results sufficiently too. The discussion is quite limited but sufficient, especially since there is no critique of the results against the literature. The discussion could be enlarged to explain the importance of foreign and national investment separately.

The conclusions can be well derived from the results and the discussion

Tables and graphs are well-developed.

Langage is correct.

References are well applied.

Recommendation: accept with minor review

 Minor editing of English language required.

Reviewer 2 Report

Report Review on the paper

" Contributions of Investment and Employment to the Agricultural GDP Growth in Egypt: An ARDL Approach"

 This paper aims to explore the impact of investment and employment on Egypt’s agriculture growth during the period 1991 to 2021 using an ARDL approach.

The paper should be improved. Below some remarks:

1-      The authors used standard unit root tests (ADF, and PP). Both tests do not allow for structural breaks. However, a major criticism of these tests is that they may suffer from power deficiency in the presence of structural breaks. Therefore, it is important to consider more recent and innovative tests that allow for structural breaks such as Zivot and Andrew (1992), Lee and Strazicich (2013, 2003) LM tests, Narayan and Popp (2010), or RALS-LM (2014). These tests are more recent and more accurate compared to the standard tests.

2-      In page 6, line 231, the authors state that “The testing techniques are expressed as follows”, but really no techniques are given below.

3-      In pages 5-6, lines 215-216, there is a mistake. The authors state that “According to Pesaran and Pesaran (1997), the best lag order is determined by the largest AIC or SBC value”. However, the best the best lag order is determined by the smallest AIC or SBC value and not largest one. Accordingly, the authors should review the table 4 and check for any necessary changes.

4-      In table 8, we observe the ARDL diagnostic tests. The authors limit the analysis to the serial correlation and heteroscedasticity. However, a more comprehensive analysis is recommended by adding tests on functional form, and on heteroscedasticity.

5-      In page 10, lines 341-342, the authors state “In order to identify the co-integration relationship for the variable under consideration, 341 we have estimated ARDL model under the case of (restricted constant and no trend) (case 2)”. Any further explanations of the choice of case 2 are recommended.

6-      The literature review suffers from various problems:

-          Some references are not cited in the text such as Abdelaal and El-Shafei 2022.

-          Studies on the international context are very limited.

-          Lack of organization and conformity to the journal requirements.

-          Lack of recent literature.

7-      Equations should be reviewed.

8-      Some perspectives for future research directions should be added.

Moderate revision

Round 2

Reviewer 2 Report

Broadly,  I'm satisfied with the corrections made.